# Pharmacological and Therapeutic Properties of *Punica granatum* Phytochemicals: Possible Roles in Breast Cancer

**DOI:** 10.3390/molecules26041054

**Published:** 2021-02-17

**Authors:** Marius Alexandru Moga, Oana Gabriela Dimienescu, Andreea Bălan, Lorena Dima, Sebastian Ionut Toma, Nicușor Florin Bîgiu, Alexandru Blidaru

**Affiliations:** 1Department of Medical and Surgical Specialities, Faculty of Medicine, Transilvania University of Brasov, 500032 Brasov, Romania; mogas@unitbv.ro (M.A.M.); dimienescu.oana.gabriela@unitbv.ro (O.G.D.); andreea.balan@unitbv.ro (A.B.); 2Department of Fundamental, Prophylactical and Clinical Disciplines, Faculty of Medicine, Transilvania University of Brasov, 500032 Brasov, Romania; lorena.dima@unitbv.ro (L.D.); toma.ionut@unitbv.ro (S.I.T.); 3Department of Surgical Oncology, Oncological Institute “Al. Trestioneanu” of Bucharest, University of Medicine and Pharmacy Carol Davila, 020021 Bucharest, Romania; alexandru.blidaru@umfcd.ro

**Keywords:** breast cancer, pomegranate, *Punica granatum*, pomegranate polyphenols

## Abstract

*Background*: Pomgranate (*Punica granatum*) represents a high source of polyphenols with great bioavailability. The role of this fruit in the prevention and treatment of various malignant pathologies has been long time cited in both scientific and non-scientific literature, making thus important to identify its involvement in the pathophysiological processes. The treatment for breast cancer had focused on the inhibition of the mechanisms that governs the estrogen activity. These mechanisms are covered either by the antagonism of the estrogen receptor (ER) or by the inhibition of the estrogen synthesis. Our interest in identifying a bioactive compound rich in polyphenols, which induces both the antagonism of the estrogen receptor, and the inhibition of the estrogen synthesis, revealed us the pomegranate fruit and its derivatives: peel and seeds. Pomegranates’ chemical composition include many biological active substances such as flavonols, flavanols, anthocyanins, proanthocyanidins, ellagitannins and gallotannins. *Materials and Methods*: We performed a review of the scientific literature by using the following keywords: “pomegranate”, “breast cancer”, “*Punica granatum*”, “pomegranate polyphenols”. Our search was performed in the PubMed and Google Scholar databases, and it included only original research written in English from the last 20 years. None of the articles were excluded due to affiliation. A total number of 28 original papers, which mentioned the beneficial activity of pomegranate against breast cancer, were selected. Both clinical and preclinical studies were considered for this review. *Results*: Recent discoveries pointed out that polyphenols from *Punica granatum* possess strong anti-cancer activity, exhibited by a variety of mechanisms, such as anti-estrogenic, anti-proliferative, anti-angiogenetic, anti-inflammatory, and anti-metastatic. Pomegranate extracts induced cell cycle arrest in the G0/G1 phase, and induced cytotoxicity in a dose- and time-dependent manner. Moreover, several polyphenols extracted from pomegranate inhibited the invasion potential, migration and viability of breast cancer cells. The effects of pomegranate juice on serum estrogens and other sexual hormones levels were also investigated on two human cohorts. *Conclusions*: *Punica granatum* represents a promising area in oncology. The large availability and low cost, associated with the lack of side effects, made from this natural product a great strategy for the management of breast cancer. There are several mechanistic studies in mouse models and in breast cancer cell lines, suggesting the possible pathways through which polyphenols from pomegranate extracts act, but larger and better-controlled studies are necessary in the future. Only two small clinical trials were conducted on humans until now, but their results are contradictory and should be considered preliminary.

## 1. Introduction

Cancer is a disease with high prevalence all over the world [1], which is mainly characterized by unrestricted proliferation of the malignant cells [2]. The origin of this pathology is still not well known, but there is a strong correlation between the genetic origin, environmental factors, epigenetic factors [3,4] and a wide range of risk factors, such as obesity, smoking, alcohol consumption, drugs, diet, treatments, etc. [5].

Breast cancer represents the second leading cause of cancer-related deaths in women of all ages [6]. Moreover, this disease shows over time a significant heterogeneity between males and females [7]. In women, between 1990 and 2017, the number of newly diagnosed cases on the global level increased from 870,000 to approximately 1,937,000, with a predominance in developing countries [7]. Early age at menarche, family history of breast cancer, late age of menopause, old age, later age at birth of the first-born child, and the long-term administration of hormone replacement therapy during menopause represent the most common risk factors for the development of breast cancer in women [8]. The administration of steroid hormones, especially estrogens, was demonstrated to play a central role in the development of this pathology [9].

Over the years, considerable advancements in breast cancer treatment options were made. However, breast cancer-related mortality continues to increase [6]. In the last years, the attention of the researchers has been focused on the prevention of this disease, as a management strategy [10]. Currently, it is estimated that 2/3 of cancer-induced deaths may be prevented through dietary changes [11]. For example, the Mediterranean diet has been reported to exert beneficial effects in the prevention of breast cancer [12]. The importance of the natural products has been recognized by centuries, and various medicinal systems promoted the use of plant-derived substances in order to prevent or treat cancer, including breast cancer [13]. These natural compounds are known as phytochemicals, and in recent years, the attention has focused on the beneficial effects of plant foods containing polyphenolic compounds [14].

Pomgranate fruit (*Punica granatum*) also possesses anticancer effects. For this reason, in the last years, pomegranates have attracted the attention of the researchers more and more. It has been shown that pomegranate modulates cellular proliferation and transformation, inflammation, and angiogenesis process [15]. Furthermore, this product and its bioactive compounds are considered to suppress the final steps of carcinogenesis and metastasis [16]. In breast cancer, pomegranate exhibits the following effects: anti-aromatase and anti-estrogenic activities [17], regulates the transforming growth factor beta (TGF-β)/Smads pathway [18], exerts anti-inflammatory effects through the reduction of pro-inflammatory cytokines/chemokines, reduces vascular endothelial growth factor (VEGF) levels [19,20,21], downregulates the expression of the genes involved in the damage of DNA [22], downregulates the estrogen-responsive genes [23], and disrupts estrogen receptor (ER) and Wnt/β-catenin signaling pathways [24]. Figure 1 is a schematic representation of the main mechanisms of *Punica granatum’s* activity against breast cancer.

In this study, we discuss the major phenolic compounds of *Punica granum* and our objective is to highlight the effectiveness of this natural product in breast cancer prevention and treatment. Despite the number of studies on the anticancer effects of pomegranate is increasing, we observed that there is a lack of human clinical trials, and we hope that this article will pave the way to future human studies on this topic.

## 2. *Punica granatum*: History and Chemical Composition

*Punica* is a genus of small trees, and its better-known species is *Punica granatum* (family Lythraceae), or pomegranate, which is native to several Asian countries including Iran to Northern India [1,25]. In the ancient Egypt, the pomegranate was used to decorate sarcophagi and it was considered a symbol of ambition and prosperity. Later, in Ayurvedic medicine, this fruit was considered “a pharmacy unto itself”, because of its multiple beneficial effects in health promotion [26]. During the years, the pomegranate was considered as a phytochemical reservoir of heuristic medicinal value. The tree and its fruits are divided in several anatomical compartments, as follows: seeds, juice, peel, leaf, flowers, bark and roots, and each of these parts are rich in bioactive molecules with therapeutic activities [15]. It has been shown that peels and juice possess strong antioxidant properties, while peel, juice and oil are weakly estrogenic and may be used for the management of postmenopausal symptoms [15]. Dried pomegranate peels have been used for the treatment of ulcers, aphthae or diarrhea. Mixtures of pomegranate seed, juice and peel products have been reported to prevent abortion [27]. Futhermore, other potential therapeutic properties of pomegranate, such as the prevention and treatment of various cardiovascular diseases, diabetes, erectile dysfunctions, Alzheimer’s disease, obesity, reproductive disorders, arthritis, etc. have been described [27].

The fruit of *Punica granatum* consists of multiple seeds, embedded in a white membrane, surrounded by the pericarp, a thick skin, which represents almost 50% of the fruit weight. Arils (40%) and seeds (10%) represent the remaining 50% of the fruit. The seeds contain approximately 20% oil and weight only 10% of the fruit weight [28]. Pomegranate is a fruit very rich in polyphenols. The concentration of polyphenols is significantly higher in pomegranate juice in comparison with other fruit juices, such as apple, orange, black cherry, or grape juice (3.8 mg/mL gallic acid vs. 0.46–2.6 mg/mL of gallic acid) [29].

The pericarp is a rich source of bioactive molecules, such as ellagitannins, polyphenols, flavonoids (luteolin, kaempferol, quercetin), and anthocyanidins (delphinidin, cyanidin, pelargonidin) [30,31,32,33] (Figure 2). Furthermore, it contains complex polysaccharides and minerals, such as sodium, potassium, calcium, phosphorus, nitrogen, and magnesium [28]. Alkaloids such as pelletierine were mentioned as an integrative part of the pomegranate peel, but their presence is still controversial [34]. Using the chromatographic method, Murthy et al. [35] revealed the presence of catechin and gallic acid in the pomegranate peels. Delphinidin has been shown to possess growth inhibitory activity in breast cancer cells. Moreover, in HER2-overexpressing breast cancer cells, this bioactive molecule was able to increase the potential of the existing targeted therapies [36].

Pomegranate juice consists of water (85.4%), polyphenols (0.2–1%), sugars (10.6%) and pectin (1.4%) [27]. The brilliant color of the pomegranate juice is provided by anthocyanins, and during the pressing process, the color declines [37]. Minor quantities of fatty acids, sterols, organic acids, and triterpenoids were also reported. Minerals in the juice include medium concentrations of calcium, sodium, magnesium, selenium, zinc, cesium, cobalt, etc. [38]. Figure 3 illustrates the chemical composition of pomegranate juice.

Pomegranate seeds contain high levels of tannins such as ellagic acid, punicalagin, gallagic acid, punicalin, etc. and anthocyanins, such as delphinidin, cyanidin, and pelargonidin [39,40] (Figure 4). The pomegranate seed oil, especially consists of fatty acids, which comprises over 95% of the oil [15]. Punicic acid, which is an isomer of linoleic acid, can be found exclusively in the pomegranate seeds and represents almost 76% of the seed oil [41]. Other minor components are represented by sterols, steroids and cerebroside. The seed matrix contains variable levels of lignins, isoflavones and hydroxybenzoic acids. The seed coat consists of organic acids such as citric acid, malic acid or ascorbic acid [40].

The leaves of the pomegranate fruit contains both unique tannins and flavones, such as apigenin [42]. They also contain chemical elements such as sodium, calcium, potassium, and iron. In young age leaves, potassium levels are high, while in old leaves calcium and iron abound. Medium-age plants have high sodium content. Furthermore, the levels of the elements also possess a seasonal variation [43,44].

The pomegranate flowers contain the same compounds also found in the peels and seeds, such as gallic acid or ursolic acid. The biochemical composition of the pomegranate flowers is still not well known and further studies are necessary in order to elucidate the distinctive compounds from this part of the pomegranate [45].

Figure 5 illustrates chemical structures of the most common bioactive molecules isolated from pomegranate.

## 3. Materials and Methods

This article is a literature review of the beneficial effects of *Punica granatum* on breast cancer prevention and treatment. Given the increasing prevalence of breast cancer around the world, and the increasing scientific proof regarding the benefits of natural compounds intake, the main objective of our research consisted of showing whether and how various extracts from the pomegranate fruit may interfere with the carcinogenesis process. We selected the relevant studies from PubMed and Google Scholar databases, using the following Medical Subject Headings (MeSH) keywords: “breast cancer”, “pomegranate”, “*Punica granatum*”, “pomegranate polyphenols”. Two authors independently identified the relevant papers and selected them following the inclusion criteria: full-text original articles written in English, including both preclinical and clinical studies. None of the articles were excluded due to affiliation. The exclusion criteria consisted of abstracts or duplicate papers and articles written in other languages than English. A total number of 28 original papers which mentioned the beneficial activity of pomegranate against breast cancer, were selected.

## 4. Results and Discussion

Strong evidence suggests that various extracts of pomegranate possess therapeutic effects against various type of cancer, including breast cancer, which represents a common cause of death in worldwide women. The mechanisms of action usually depends on the molecular subtypes of breast cancer, but the main common pathways interfere with the mutation process, proliferation of the malignant cells, invasion, inflammation, angiogenesis, and metastasis [46]. The most significant therapeutically active polyphenols from pomegranate are ellagic acid, punicic acid, ellagitannins, anthocyanins and anthocyanidins, flavones, flavonoids and estrogenic flavonols [47].

The chemotherapeutic drugs currently used for the treatment of breast cancer have several side effects, and a major limitation of the conventional treatment consists of multidrug resistance [48]. For this reason, researchers have isolated from pomegranate peel or pericarp several phytochemicals with similar anticancer effects. These phytochemicals could be translated in the future into marketable anti-breast cancer drugs [49]. Moreover, some of these extracts can also potent the action of chemopreventive drugs, such as tamoxifen [50].

Estrogen is a pleiotropic hormone, which exerts many actions in the reproductive organs, such as uterus, ovary and breast. Moreover, it also acts on non-reproductive tissues including bone, the central nervous system and the cardiovascular system [51]. Cumulative exposure of the breast epithelium to high levels of estrogen represents the main risk factor for the development of breast cancer [52]. The best strategy for the prevention and treatment of estrogen-dependent breast cancer is to selectively block its activity in the affected tissues. Tamoxifen is the most well-known synthetic antiestrogen used in the treatment of ER+ breast cancer. Despite it has beneficial effects in breast cancer, its agonism on the uterine endometrium is leading to a questionable connection to endometrial carcinoma [53]. Estrogens interact with two different members of the nuclear receptor, known as ERα and ERβ selective estrogen receptor modulators (SERMs) are agents used in the conventional hormonal therapy for estrogen-dependent breast cancers, which bind ERs and exert agonist or antagonist functions, depending on the type of tissue on which they act [54]. The genes regulated by SERMs with ERα are distinct from those regulated by ERβ [55]. In these conditions, it is stated that drugs targeted selectively to ERα or ERβ will produce more selective clinical effects. Increasing evidence have shown that ERα promotes the proliferation of breast cancer cells, while ERβ acts as a tumor suppressor. Therefore, ERα-selective antagonists might be effective in the prevention and treatment of estrogen-dependent breast cancer [56]. ERα antagonists have shown increase resistance in treating estrogen-dependent breast cancer for a long period. For this reason, recent studies suggested the importance of studying the role of ERβ in the treatment of estrogen-dependent breast tumors. Genistein is an isoflavone extracted from *Punica granatum* and it may act on both ERα and ERβ [57]. According to Pons et al. [58] the treatment with genistein induced cell cycle arrest in T47D breast cancer cells with a low ERα/ERβ ratio, and also improved the mitochondrial functionality. In MCF-7 breast cancer cells with high ERα/ERβ ratio and in ER- MDA-MB-231 cells, these effects have not been observed.

Methanol extracts of pomegranate pericarp were demonstrated to possess SERM properties [23]. Methanol extract of pomegranate pericarp significantly inhibited the proliferation of ER+ MCF-7 cells, in a dose-depending manner, and did not influence the proliferation of ER- MDA MB-231 cells. Moreover, the pomegranate pericarp extract inhibited 17β-estradiol-induced proliferation and growth in ER+ MCF-7 cell lines. In ovariectomized mice, the pomegranate pericarp extract did not increase the weight of the uterus.

In has also been demonstrated that *Punica granatum* exhibits SERM-like properties in organs other than breast. Cardioprotective effects of the pericarp extracts were comparable to that of 17β-estradiol [23]. Moreover, the protective effects of pomegranate in the brain have been mentioned. According to Valdes-Sustaita et al. [59] the aqueous extract of pomegranate administered by the oral route induced antidepressant-like actions in ovariectomized mice. These actions were mediated by ERβ and by the serotonergic system. Mori-Okamoto et al. [60] have also shown that pomegranate supplementation may exert estrogen-like effects in ovariectomized mice by acting against the depressive state and bone loss.

In the ovarian granulosa cells, aromatase enzyme converts androstenedione to estrone (E1). Estrone possesses weak estrogenic activity, but it can be reduced to estradiol (E2), which is much more active in breast tissue, by the enzyme 17β-hydroxysteroid dehydrogenase [61]. Thus, the products that reduce estrogen synthesis through the inhibition of this key enzyme activity have the potential to lead to effective prevention and treatment strategies for breast cancer. Kim et al. [62] pointed out that polyphenols from whole pomegranate seed oil, fermented pomegranate juice and aqueous pericarp extract were able to inhibit the activity of 17-β-hydroxysteroid dehydrogenase from 34% to 79%. Regarding the anti-estrogenic effects, polyphenols from both pericarp and fermented pomegranates juice inhibited the growth of ER+ MCF-7 and ER- MB-MDA-231 breast cancer cell lines. Furthermore, lyophilized fresh pomegranates juice inhibited the estrogenic activity of 17-β-estradiol by 55%. In ER+ MCF-7 breast cancer cells, pomegranate seed oil also inhibited the invasion process and induced apoptosis in ER- MDA-MB-435 metastatic human breast cancer cells. The researchers have found a stronger potency of the fermented juice polyphenols, in comparison with pericarp polyphenols. This fact could be attributed to the breakage of flavonoid-sugar complexes during fermentation, which is similar with the stomach hydrolysis.

Adams et al. [17] also conducted a study that pointed out anti-estrogenic effect of bioactive molecules extracted from pomegranates. Ten ellagitannin-derived compounds (urolithin A, urolithin B, gallagic acid, ellagic acid, etc) were isolated from pomegranates and used to investigate their anti-aromatase effects. Urolithin B exhibited the most significant anti-aromatase activity. Moreover, this compound inhibited the proliferation on ER+ MC7-aro cells, induced by testosterone. Six out of 10 compounds exhibited anti-aromatase activity, and all exhibited antiproliferative effects, but the most efficient was urolithin B, followed by gallagic acid.

In order to investigate the effects of a pomegranate emulsion on intratumoral expression of ER-α, ER-β, β-catenin and cyclin D1 in a mouse model, Mandal et al. [24] used immunohistochemical techniques to determine the amount of these molecules. They observed that pomegranate emulsion decreased the independent expression of intratumor ER-α and ER-β and the ER-α:ER-β ratio. Moreover, nuclear translocation of β-catenin and its cytoplasmatic accumulation were also reduced. Through these mechanisms, pomegranate emulsion induced anti-proliferative and pro-apoptotic effects in tumoral cells cultures. Bishayee et al. [63] also demonstrated the chemopreventive effects of pomegranate emulsion on rat mammary tumors induced by dimethylbenz(a)anthracene (DMBA). After the treatment of the tumoral cells with pomegranate emulsion, cell proliferation was significantly decreased and histopathological changes were reversed. Moreover, the pomegranate emulsion increased the expression of Bax gene, decreased Bcl2 and up-regulated the caspase cascades.

Despite the evidence of its anti-estrogenic activities, there is only a small amount of data suggesting the possible estrogenic action of pomegranates. Recent studies have shown that ellagic acid may exert both estrogenic and anti-estrogenic activity, depending on the estrogen receptor to which it binds [49]. Sharaf et al. [64] injected pomegranate seed oil in ovariectomized mice and observed increased cornification of the vaginal cells. Moreover, ellagic acid exerted strong estrogenic effects in osteoblastic cell lines, while in MCF-7 cell lines demonstrated potent anti-estrogenic activity. According to Papoutsi et al., the binding of ellagic acid to the estrogen-alpha receptor (ERα) induces estrogenic effects, whereas the binding to ERβ led to anti-estrogenic activity [65].

Strati et al. [66] also investigated both estrogenic and anti-estrogenic effects of ellagic acid in breast cancer. Their main objective was to evaluate the potential of this compound to inhibit the expression of human telomerase reverse transcriptase (hTERT) α+β+ splice variant in MCF-7 breast cancer cells. Telomerase is a promising tumor maker whose role in the oncogenic transformation has been intensively studied in the recent period. The malignant transformation of normal human breast cells in vitro was achieved by reconstitution of telomerase activity by induction of hTERT gene expression in combination with other oncogenes. The authors investigated the possibility that ellagic acid possesses antibreast cancer properties via modulation of hTERT α+β+ mRNA expression. Their results indicated that ellagic acid induced hTERT α+β+ mRNA expression in the absence of 17β-estradiol, while the coexistence of ellagic acid with this molecule inhibited the hTERT α+β+ mRNA expression. These findings suggest that ellagic acid may exert both estrogen agonistic effects and antagonistic effects on the expression of hTERT α+β+ mRNA in breast cancer cells, through an ER dependent mechanism.

Evidence of the free radical scavenging property of pomegranates extracts have also been published. Several studies have pointed out that cold-pressed pomegranate seed oil and fermented pomegranates juice exerted strong antioxidant activity [67]. According to Schubert et al. [67] cold pressed seed oil showed the inhibition of cyclooxygenase (COX2) by 31–44% and inhibition of lipoxygenase by 69–81%. COX2 is overexpressed in breast cancer and it is also involved in metastasis process of breast cancer [68]. The downregulation of this molecule can cause the downregulation of aromatase enzyme, through PGE2 pathway [69]. Shukla et al. [70] measured the plasma levels of COX2 in rabbits, at 2 h after the ingestion of a polyphenol rich extract of pomegranate. They observed that the activity of COX2 was significantly reduced by the pomegranate extract.

Chronic inflammation is another risk factor for the malignant transformation. This process also aids the growth of malignant cells, angiogenesis, invasion and metastasis. In these conditions, the suppression of pro-inflammatory cytokines and chemokines might reduce the survival rate of the tumoral cells [46]. Studies have shown that pomegranate extracts also possess anti-inflammatory effects. Mandal et al. [71] analyzed the anti-inflammatory mechanism of orally administered pomegranate emulsion in Sprague-Dawley rats with DMBA-induced mammary tumorigenesis. They observed that pomegranate emulsion decreased the expression on COX2 and heat shock protein 90 (HSP90), and decreased the translocation of NF-κB from cytosol to nucleus. Furthermore, pomegranate emulsion exerted chemopreventive effect by enhancing apoptosis and reducing cell proliferation. Another study evaluated the anti-inflammatory potential of the hydrophilic fraction (80% aqueous methanol extract) of pomegranate seed oil. Constantini et al. [19] have showed that several pro-inflammatory cytokines and VEGF levels significantly decreased in both ER+ MCF-7 and ER- MDA MB-231 breast cancer cell lines, after the treatment with a hydrophilic fraction of pomegranate seed oil. The targeted cytokines were IL-2, 6, 12, 17, MCP-1, MIP-1α/1β, and TNF-α. Furthermore, punicic acid and its congeners decreased the viability in the studied cells. Ellagic acid, ursolic acid and luteolin from pomegranates also reduced the proliferation and induced apoptosis in a mouse mammary cancer cell line (WA4). According to Dai et al. [72] a standardized extract of pomegranate arrested cell cycle progression in the G0/G1 phase leading to the inhibition of cell proliferation.

Shirode et al. [22] performed a study based on the anti-cancer properties of pomegranate extracts, and focused on the genomic changes occurred in ER+ MCF-7 breast cancer cell lines, after the administration of pomegranate extracts. The authors reported that after the treatment, 398 genes were downregulated, while 505 genes were upregulated. The upregulated genes were involved in the induction of apoptosis, and cell proliferation. On the other side, the downregulated genes included those genes involved in the chromosomal organization, mitosis, DNA damage response and repair, and RNA processing. Pomegranate extract downregulated the genes involved in DNA double strand break repair, such as BRCA1, BRCA2, BRCC3, RAD 50 and 51, NBS1. Furthermore, pomegranate extract regulated miRNAs involved in DNA repair processes.

A recent study performed by Pan et al. [73] investigated the effects of punicalagin, a bioactive compound of pomegranate, on ER+ MCF-7 and ER- MDA MB-231 breast cancer cells. Golgi phosphoprotein 3 (GOLPH3) was transfected into both treated and non-treated cells, and its levels were examined using quantitative real-time polymerase chain reaction (qRT-PCR). Doses of 50 μM of punicalagin inhibited the migration, invasion and viability of both ER+ and ER- cells. Moreover, punicalagin decreased the expression of MMP-2, MMP-9, GOLPH3, and N-cadherin, and upregulated E-cadherin. Through these mechanisms, punicalagin may suppress cell viability and may decrease the metastasis process.

Of all the processes involved in breast cancer carcinogenesis, metastasis represents one of the most difficult to target. However, evidence from the literature have showed that pomegranate contains various bioactive molecules capable to suppress the tumor cell invasion. It has been demonstrated that nuclear transcription factor-kB (NF-kB) regulates the inflammatory process, tumorigenesis, cell survival and cell proliferation. According to Khan et al. [74], pomegranate fruit extracts consisting of both seed oil and fermented juice may exert anti-cancer activity by modulating the NF-kB pathway. The authors used aggressive breast cancer cell lines (ER-, PR- MDA MB-231 and ER-, PR- SUM149) and examined the effects of pomegranate extracts on cell survival, proliferation and invasion, these processes being regulated via NF-kB pathway. The main bioactive constituents from the aqueous extract were ellagitannins and phenolic acids, and in the lipid phase of pomegranate fruit extract predominated conjugated octadecatrienoic acids. The aqueous extract of pomegranate fruit inhibited the proliferation and invasion rates in the cell lines, by modulating NF-kB pathway. Moreover, the motility of the aggressive breast cancer cell lines was significantly decreased by suppressing RhoA and RhoC protein expression. The crosstalk between RhoC and NF-kB interaction has been also suggested. Therefore, according to the results of this study, we can conclude that antimetastatic potential of pomegranate fruit extracts is not dependent by the presence of ER or PR.

Mehta et al. [75] investigated the chemopreventive activity of a purified chromatographic peak of pomegranate fermented juice polyphenols and of whole pomegranate seed oil, in a mouse mammary organ culture. For the first 10 days of culture, the mouse mammary glands were treated with a high-performance liquid chromatographic peak separated from pomegranate juice, or pomegranate seed oil, and on day 3, the glands were exposed to a carcinogen. After 10 days, the pomegranate fermented juice polyphenols induced a 42% reduction in the number of lesions, while the purified chromatographic peak of pomegranate fermented juice polyphenols and pomegranate seed oil each induced an 87% reduction of the lesions. This study highlighted the preventive potential of pomegranate against breast tumors.

Another study conducted by Rocha et al. [76] have demonstrated that pomegranate juice and three of its major bioactive molecules (luteolin, punicic acid and ellagic acid) were able to inhibit the processes involved in breast cancer metastasis. Pomegranate juice or its combined components inhibited the growth of ER+ MCF-7 and ER- MDA MB-231 cell lines. Furthermore, pomegranate juice, luteolin, ellagic acid and punicic acid increased tumor cell adhesion and degreased cell migration, without affecting the non-neoplastic breast cancer cell line (MCF10A). Another effect of the pomegranate juice and its compounds was the inhibition of SDFF1 α, a chemokine that in normal conditions attracts breast cancer cells to the bone and stimulates the metastatic process. Regarding the inflammatory process, Rocha et al. reported that pomegranate juice and its major phenolic compounds also decreased inflammation in breast cancer cell lines, negatively influencing the progression of carcinogenesis. All the investigations performed until now highlighted the relevance of pomegranate extracts and its bioactive molecules in promoting migration and invasion of breast cancer cells, by various mechanisms.

Punicic acid is an omega-5 long chain polyunsaturated fatty acid extracted from pomegranate seeds oil, whose proliferation inhibitory effects were also studied by Grossmann et al. [21]. They showed that punicic acid was able to inhibit the proliferation in MDA-MB-231 and MDA-ERα7 cells by 92% and 96%, respectively. Furthermore, this polyphenol induced apoptosis and disrupted cellular mitochondrial membrane potential in the targeted tumoral cells.

Angiogenesis, defined as the development of novel blood vessels, is essential for the progression of carcinogenesis and for metastasis. Furthermore, the number of new developed blood vessels is one of the deciding factors for the prognosis [77]. Several studies indicated that pomegranate extracts possess the ability to inhibit angiogenesis. The treatment with fermented pomegranate juice and supercritical CO_2_-extracted pomegranate seed oil significantly decreased proangiogenic VEGF in ER+ MCF-7 cell lines, and less in ER- MDA MB-231 cell lines. Pressed pomegranate seed oil and pomegranate peel extract exerted mild effects on VEGF levels. Furthermore, the same study of Toi et al. [77] reported that the antiangiogenic migration inhibitory factor (MIF) was significantly increased in ER- MDA MB-231 cells after the treatment with fermented pomegranate juice and supercritical CO_2_-extracted pomegranate seed oil. Survivin is a proteic molecule, which controls angiogenesis, cell proliferation and apoptosis, and is overexpressed in breast cancer [78]. According to Barnejee et al. [50], pomegranate downregulates SP transcription factors, along with survivin, VEFG, and VEGFR, which are regulated by SP factors.

Inhibition of cells proliferation and induction of apoptosis represent two of the main mechanisms of *Punica granatum* against breast cancer. Jeune et al. [79] cultured MCF-7 cancer cells for 48 h and then exposed to pomegranate extracts and genistein for 24 h, in both single and combined treatments. After the exposure, the researchers observed that both genistein and pomegranate extracts exerted significant dose-dependent and time-dependent cytotoxicity, growth inhibition and pro-apoptotic effects. Moreover, combined treatments were more efficient in comparison with single treatments.

Another study that focused on the inhibition of cellular proliferation induced by polyphenolic compounds of pomegranate was conducted by Tanner et al. [80]. They used MDA-MB-231 tumor cells and exposed the cultures to pomegranate juices with different polyphenolic content. Cellular proliferation of the ER- breast tumor cells was inhibited by pomegranate juice in a dose- and time-dependent manner, and the response was comparable to that of high-dose cisplatin, showing the high efficiency of pomegranate polyphenols against breast cancer cells.

Multidrug resistance is one major hindrance to chemotherapy in breast cancer, and during the years, the researches have tried to overcome this problem, in order to make chemotherapy more effective. Banerjee et al. [50] aimed to determine the effect of pomegranate extracts on both sensitive and tamoxifen (TAM)-resistant-MCF-7 cell lines. Their results have showed that pomegranate extract decreased cell viability and induced cell death, enhancing the action of tamoxifen in both TAM-resistant MCF-7 and sensitive breast cancer cells. Quercetin is a component of pomegranate juice and it can be also found in pomegranate pericarp. According to Scambia et al. [81], quercetin was able to decrease adriamycin (ADR) resistance in ER+ MCF-4 ADR-resistant breast cancer cell line, in a dose-dependent manner. Moreover, kaempherol, which is present in the pericarp of pomegranates, might increase the concentration of the chemotherapy drug into the tumor cells, by inhibiting the pump responsible for the drug efflux [82].

Despite the documented beneficial effects of pomegranate polyphenols, their poor absorption, short retention time and low systemic bioavailability, may decrease their chemopreventive potential [83]. For this reason, the encapsulation of pomegranate ellagitannins using biodegradable and biocompatible nanoparticles may skip the gastrointestinal hydrolysis and increase their systemic bioavailability [84]. Nanotechnology approaches were initially used in cancer management, in order to increase bioavailability of the drugs, to decrease toxicity and to promote selective tumor uptake. A study of Shirode et al. [85] reported that nanoencapsulated pomegranate polyphenols exerted potent antiproliferative effects on ER+ MCF-7 and Hs578T breast cancer cells. Moreover, ellagic acid nanoprototypes exerted a 2- to 12-fold enhanced effect on cell growth inhibition, in comparison with free pomegranate polyphenols. Fluorescent microscopy revealed that nanoparticles of pomegranate polyphenols were efficiently taken up, and the uptake reached no more than 24 h. In conclusion, nanotechnology-enabled delivery of pomegranate bioactive compounds might be a promising strategy for the prevention and management of breast cancer.

Regarding the human evidence of pomegranate efficiency as natural anticancer agent, only two studies were conducted until present, but their results are contradictory and controversial. Kapoor et al. [86] aimed to determine the effect of pomegranate juice consumption on serum levels of estrone, estradiol, androstenedione, testosterone and sex hormone binding globulin (SHBG). They selected 64 normal weight healthy women, which were randomly assigned to drink either 8 ounces of pomegranate juice or 8 ounces of apple juice daily, during a period of 3 weeks. After this period, the serum levels of sexual hormones were measured. The investigators observed that women in the intervention group compared to control group had a significant decline in estrone and testosterone levels, but no differences were noticed in SHBG levels. However, the results should be considered preliminary and larger trials including both overweight and obese women should be conducted in order to sustain the effects of pomegranate juice on estrogen levels.

Another clinical trial that included 11 post-menopausal women, with increased risk to develop breast cancer was mentioned by Warren et al. [87]. After a period of 7 days of pomegranate juice administration (8 ounces per day), the researchers reported a statistically significant increase in serum levels of E1. However, the serum levels of E2 recorded no change. Considering that increased levels of E1 have been associated with a high risk of postmenopausal breast cancer development, this study is the only exception that does not support the consumption of pomegranate juice to prevent the development of breast cancer.

However, larger and better-designed human clinical trials are necessary to be conducted in order to establish how and whether pomegranate juice may aid to prevent breast cancer or to treat it, in association with chemotherapy drugs. Table 1 synthesizes in vivo and in vitro studies regarding the anti-breast cancer activity of *Punica granatum.*

## 5. Current Limitations and Future Perspectives of *Punica granatum* in Breast Cancer Treatment

Pomegranate fruits, leaves, flowers, and seeds have all been used in traditional herbal medicine for treating various illness, including cancer. A large amount of preclinical studies has demonstrated the beneficial effects of pomegranate bioactive molecules against breast cancer. However, human clinical trials are still lacking, this fact being a major limitation in the utilization of pomegranate as an anticancer agent. In the literature, there are only two clinical trials that investigated the effects of pomegranate on the hormonal panel and on breast cancer. The small interventional period, the low number of enrolled subjects and the lack of subjects’ heterogeneity represented, in our opinion, the main limitations of these studies.

The metabolic enzymes and pathways that confer the structural diversity of pomegranate phytochemicals also warrant further investigation in order to better understand their role in the fight against breast cancer. Moreover, the possibility of pomegranate polyphenols to act as Pan Assay Interference Compounds (PAINS) should be a topic of major interest for future studies. To our knowledge, no study on this field has been conducted yet.

As we previously mentioned, pomegranate extracts possess anti-aromatase effects, which represent a major anti-breast cancer mechanism. As the brain is also equipped with abundant estrogen-producing enzyme, aromatase, and neuron-derived estrogen is essential for the memory formation, could any pomegranate extract to influence the brain aromatase or could it impair the memory? This aspect should be very interesting to investigate the more so as pomegranate was demonstrated to exert neuroprotective effects by other mechanisms that not interfere with the aromatase enzyme.

## 6. Conclusions

In the current setting, breast cancer prevention and treatment via natural products such as *Punica granatum* represent a promising area of oncology. In the last period, this aspect has drawn the attention of both clinical physicians and scientists due to the ability of dietary agents to prevent and suppress the carcinogenesis process. Their large availability and low cost, associated with the lack of side effects, made from these natural products a great strategy for the management of breast cancer. A wide range of in vitro and in vivo studies pointed out that the whole pomegranate fruit, including seeds, juice, pericarp, and seeds oil possess strong anti-proliferative, anti-inflammatory, anti-tumorigenic, pro-apoptotic, anti-estrogenic and antioxidant effects. The molecular subtypes of cancer, genomic aberrations and pathways may influence the results of the treatment. There are several mechanistic studies in mouse models and in breast cancer cell lines suggesting the possible pathways through which polyphenols from pomegranate extracts act, but larger and better-controlled studies are necessary in the future. Only two small clinical trials were conducted on humans, in order to establish the effects of pomegranate juice on estrogen levels but their results are contradictory and because of several study limitations (low number of subjects, short interventional period, etc.), the results should be considered preliminary and should pave the way to further investigations in the field. In our opinion, clarifying the effects of pomegranate compounds on mammary tumors could result in important piece of information for the consumers, and could elucidate the possible effects of dietary agents on the prevention or management of breast cancer.

## Figures and Tables

**Figure 1 molecules-26-01054-f001:**
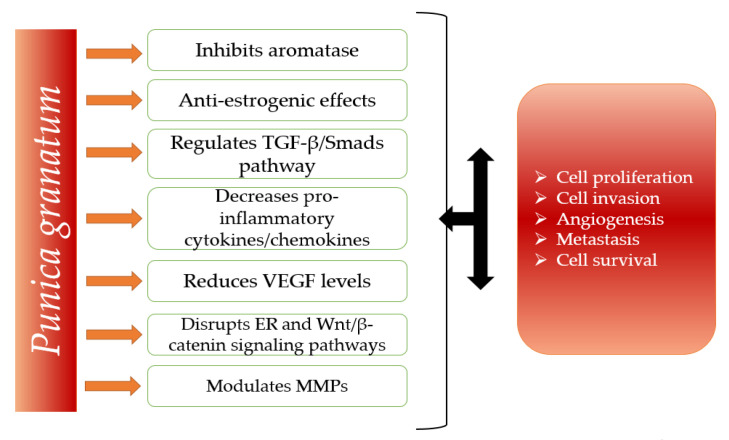
Anti-cancer mechanisms of *Punica granatum* and molecular targets.

**Figure 2 molecules-26-01054-f002:**
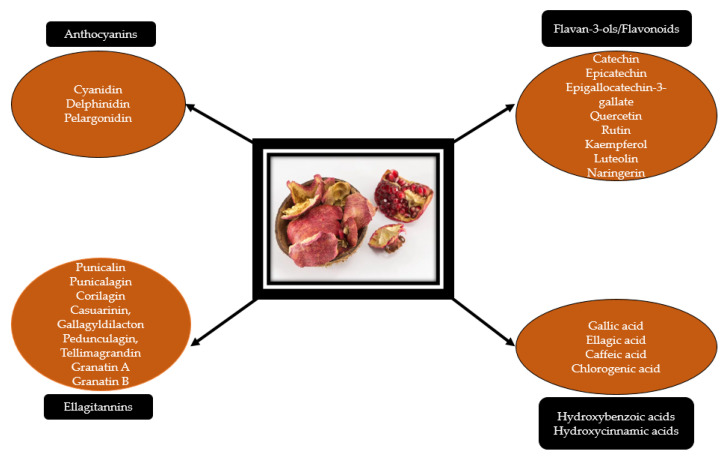
Chemical composition of pomegranate peels.

**Figure 3 molecules-26-01054-f003:**
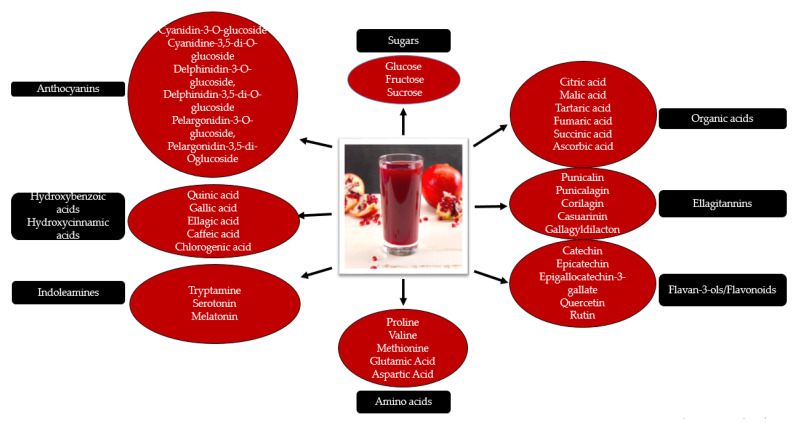
Chemical composition of pomegranate juice.

**Figure 4 molecules-26-01054-f004:**
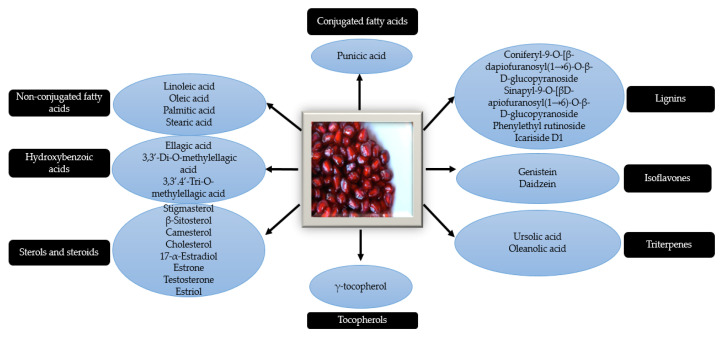
Chemical composition of the pomegranate seeds.

**Figure 5 molecules-26-01054-f005:**
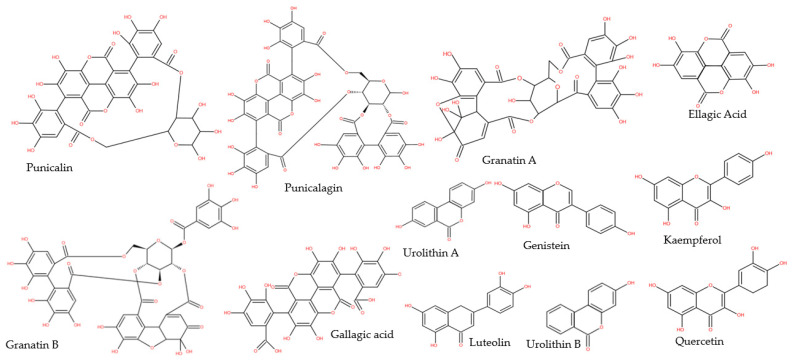
Chemical structures of the most common bioactive molecules isolated from pomegranate.

**Table 1 molecules-26-01054-t001:** Studies regarding anti-breast cancer activity of *Punica granatum.*

Author, Year, Reference	Type of Study	Pomegranate Extract/Pomegranate Polyphenol(s)	Cells Type/Subjects	Anti-Cancer Activity	Results
Kim et al., 2002 [62]	In vitro	Aqueous pericarp extract, fermented juice and cold-pressed seed oil, all rich in polyphenols	MCF-7 cells MCF-10A cells MB-MDA-231cells	Estrogenic activity Anti-proliferative Anti-aromatase	Polyphenols from fermented juice, pericarp, and oil: inhibited aromatase activity (60–80%) and inhibited 17-β-hydroxysteroid dehydrogenase type 1 from 34 to 79% Lyophilized fresh pomegranate juice inhibited 17-β-estradiol by 55% Fermented pomegranate juice polyphenols exhibited twice anti-proliferative effect than fresh juice Seed oil polyphenols inhibited the proliferation of MCF-7 cells by 90%, the invasion by 75% and induced apoptosis in MDA-MB-435 ER- metastatic human breast cancer cells by 54%
Toi et al., 2003 [77]	In vitro	Polyphenols from pomegranate fermented juice and seed oil	MCF-7 cells MDA-MB-231 cells MCF-10A cells	Anti-angiogenic effects	Both pomegranate fractions down-regulated VEGF in MCF-10A and MCF-7 MIF was up-regulated in MDA-MB-231
Mehta et al., 2004 [75]	In vitro	Polyphenols from pomegranate fermented juice and seed oil	Mouse mammary organ culture	Chemoprevention	The number of tumoral lesions was reduced with 42% by pomegranate juice polyphenols Pomegranate seed oil induced a reduction of 87% in the number of lesions
Jeune et al., 2005 [79]	In vitro	Pomegranate extracts and genistein	MCF-7 cells	Anti-proliferative effects Pro-apoptotic effects	The inhibition of cell growth and cytotoxic effects were more significant after combined treatments than after single treatments
Tanner et al., 2008 [80]	In vitro	Polyphenols from pomegranate juice	MDA-MB-231 cells	Anti-proliferative effects	Cellular proliferation was inhibited in a dose-dependent manner, and the result was comparable to that of Cisplatin
Khan et al., 2009 [74]	In vitro	Aqueous pomegranate fruit extracts	Aggressive breast cancer cell lines	Anti-metastatic effects	NF-kB—dependent reporter gene expression was reduced in aggressive breast cancer cells, in a dose-dependent manner The aqueous pomegranate fruit extracts decreased the expression of RhoC and RhoA proteins.
Strati et al., 2009 [66]	In vitro	Ellagic acid	MCF-7 cells	Chemopreventive effects	Ellagic acid down-regulated 17β-estradiol-induced hTERT α+β+ mRNA expression in breast cancer cells
Grossmann et al., 2010 [21]	In vitro	Punicic acid	MDA-MB-231cells MDA-ERα7 cells	Anti-proliferative effects Pro-apoptotic effects	Punicic acid inhibited cellular proliferation by 92% and 96% in MDA-MB-231 and MDA-ER α7, respectively Punicic acid induced apoptosis by 86% and 91% in MDA-MB-231 and MDA- ER α7, respectively Punicic acid effects were dependent on the PKC pathway and on lipid peroxidation.
Adams et al., 2010 [17]	In vitro	Pomegranate Ellagitannins (ellagic acid, gallagic acid, urolithins A, urolithins B)	MCF-7 cells	Anti-aromatase activity Anti-proliferative effects	Urolithin B showed the most potent anti-aromatase activity in breast cancer cells Urolithin B significantly inhibited testosterone-induced MCF-7 cells proliferation
Dai et al., 2010 [72]	In vitro	Standardized extract of pomegranate	WA4 cells derived from mouse MMTV-Wnt-1 mammary tumors	Anti-proliferative effects Cytotoxic effects Pro-apoptotic effects	Pomegranate extract induced the cell cycle arrest in the G0/G1 phase Pomegranate extract induced cytotoxicity in WA4 cells (concentrations >10 µg/mL) Caspase-3 enzyme activity was increased in WA4 cell line Luteolin, ellagic acid, ursolic acid induced cytotoxicity in a dose- and time-dependent manner
Dikmen et al., 2011 [20]	In vitro	Methanolic pomegranate fruit peel extract	MCF-7 cells	Anti-proliferative effects Pro-apoptotic effects Antioxidant effects	Cellular proliferation decreased in a dose- and time-dependent manner The expression of pro-apoptotic gene Bax significantly increased, after 48 and 72 h of treatment.
Banerjee et al., 2011 [50]	In vitro	Pomegranate fruit extracts	Tamoxifen resistant MCF-7 cells	Sensitizes the effects of tamoxifen	Pomegranate extract enhanced the action of tamoxifen in both sensitive and tamoxifen-resistant MCF-7 cells, by inducing pro-apoptotic effects
Wang et al., 2012 [88]	In vitro	Luteolin	IGF-1-stimulated MCF-7 cells	Anti-proliferative effects Pro-apoptotic effects	Akt phosphorylation and IGF-1R were significantly decreased The expression of ERα was significantly increased
Sreeja et al., 2012 [23]	In vitro	Methanol extract of pericarp of pomegranate	MCF-7 cells MDA-MB-231 cells	Anti-estrogenic effects	The binding of E2 to ER was inhibited The growth and proliferation of MCF-7 cells was inhibited
Barnejee et al., 2012 [89]	In vitro	Pomegranate extract polyphenols	MCF-7 cells BT-474 cells MDA-MB-231 cells MCF-10F cells MCF-12F cells	Anti-inflammatory effects Cytotoxic effects	The level of Sp proteins and Sp-regulated genes were decreased Pomegranate extracts induced SHIP-1 expression, down-regulated miRNA-155 and inhibited PI3K-dependent phosphorylation of AKT
Rocha et al., 2012 [76]	In vitro	Pomegranate juice polyphenols (luteolin, ellagic acid, punicic acid)	MCF-7 cells MDB-MB-231 cells MCF10A cells	Increase cell adhesion Anti-metastatic effects Anti-inflammatory effects	The chemotaxis of MCF-7 cells to SDF1α, a chemokine that attracts breast cancer cells to the bone, was inhibited The expression of pro-inflammatory cytokines/chemokines was significantly decreased
Shirode et al., 2013 [22]	In vitro	Pomegranate extract	MCF-7 cells	Anti-proliferative effects Pro-apoptotic effects Antioxidant effects	Pomegranate extract: induced cell cycle arrest in G2/M and apoptosis in MCF-7 cells regulated miRNAs involved in DNA replication and repair increased the frequency of DNA double strand break repair
Pons et al., 2013 [58]	In vitro	Genistein	T47D cells (low ERα/ERβ ratio) MDA-MB-231 cells (ER-) MCF-7 cells (high ERα/ERβ ratio)	Anti-proliferative effects Pro-apoptotic effects	Genistein treatment produced an up-regulation of ERβ and increased the activity of cytochrome c oxidase in T47D cells Cell cycle arrest and improved mitochondrial functionality in T47D cells E2 and genistein induced cell proliferation and apoptosis inhibition in MCF-7 cells
Costantini et al., 2014 [19]	In vitro	80% aqueous methanol extract containing conjugated linolenic acids (punicic acid)	MCF-7 cells MDA-MB-231 cells	Anti-inflammatory effects Antioxidant effects Cytotoxic effects	Punicic acid induced cytotoxicity in breast cancer tumor cells The levels of VEGF and other pro-inflammatory cytokines were decreased in a dose-dependent manner
Kapoor et al., 2015 [86]	Randomized placebo- controlled clinical trial	Pomegranate juice	38 healthy postmenopausal women at risk for breast cancer development	Induces various changes on hormonal biomarkers of breast cancer risk	The group which consumed pomegranate juice for 3 weeks registered significant decrease of estrone and testosterone levels No differences were observed in SHBG levels
Shirode et al., 2015 [85]	In vitro	Nanoparticles loaded with pomegranate extract	MCF-7 cells	Cell growth inhibition	Nanoparticles loaded with pomegranate extract inhibited the growth of MCF-7 cells, with an 2 to 12 fold enhanced effect than free pomegranate extract
Chen et al., 2015 [18]	In vitro	Ellagic acid	MCF-7 cells	Anti-proliferative effects Pro-apoptotic effects	Ellagic acid arrested cell cycle in the G0/G1 phase through TGF-β/Smads signaling pathway
Bishayee et al., 2015 [63]	In vivo	Pomegranate emulsion	Rat mammary tumors induced by 7,12-dimethylbenz(a)anthracene (DMBA)	Chemopreventive effects	Mammary tumor incidence and cell proliferation were reduced and histopathological changes were reversed Increased the expression of Bax gene, decreased Bcl2 and up-regulated caspase cascades.
Mandal et al., 2015 [24]	In vitro	Pomegranate emulsion	Rat mammary tumors induced by 7,12-dimethylbenz(a)anthracene (DMBA)	Anti-proliferative effects Pro-apoptotic effects	Downregulated the expression of ER α, β and decreased nuclear translocation of β-catenin. Decreased the expression of protein cyclin D1, a cell growth regulatory molecule
Mandal et al., 2017 [71]	In vivo	Pomegranate emulsion	Rat mammary tumors	Anti-inflammatory effects Anti-proliferative effects Pro-apoptotic effects	Decreased the levels of HSP90, COX-2 The translocation of NF-kB to the nucleus was hindered by the pomegranate emulsion
Nallanthighal et al., 2017 [90]	In vitro	Pomegranate extract	Breast cancer stem cells (CSCs): neoplastic mammary epithelial (HMLER) and Hs578T	Chemopreventive effects	Inhibited the growth and self-renew ability of CSCs and inhibited their ability to migrate
Pan et al., 2020 [73]	In vitro	Punicalagin	MCF-7 cells MDB-MB-231 cells	Pro-apoptotic effects Anti-metastatic effects	50 μM or higher doses inhibited the invasion potential, migration and viability of breast cancer cells. Decreased the expression of N-Cadherin, GOLPH3, MMP-2 and MMP-9

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
