# Peer review of "Pharmacological and Therapeutic Properties of Punica granatum Phytochemicals: Possible Roles in Breast Cancer"

_molecules, 2021, doi:10.3390/molecules26041054_

Round 1

Reviewer 1 Report

This is a well written review paper. The authors comprehensively discussed the chemical components of pomegranate, the anti-cancer effects, as well as the possible mechanisms based on current studies. They further briefly discussed potential limitations of pomegranate polyphenols application. As pomegranate is an easily available fruit in daily life, unveiling the anti-cancer effects of pomegranate supplementation will greatly boost the economic treatment of breast-cancer in the future. My comments are listed below: 

  1. It’s well known that neurons in the brain are also equipped with abundant estrogen-producing enzyme, aromatase, and neuron-derived estrogen is essential for memory formation. Since Punica granatum has potent anti-estrogenic effects, will Punica granatum treatment also suppress neuronal estrogen synthesis and thus impair cognitive functions? Please discuss.
  2. Is Punica granatum also a selective estrogen modulator, like Tamoxifen which exerts distinct estrogen regulatory actions in different organs? If so, please provide evidences.
  3. This paper concludes that Punica granatum has anti-cancer activity by a variety of mechanisms, like anti-estrogen, anti-proliferation, anti-angiogenesis, anti-inflammation, anti-metastasis et al, which is shown in Table 1. It will be more informative if the authors can summarize all these mechanisms of Punica granatum’s anti-breast cancer effects with schematic illustration in a figure.
  4. Since in vivo animal studies, especially clinical studies are still lacking, it will be important if the authors can make a separate section to discuss the current limitation and future direction of Punica granatum treatment in breast cancer therapy.

Author Response

Dear Reviewer,

We thank you for your cooperation and we appreciate you taking the time to analyze our work. Following your comments and suggestions, we revised our paper, in order to have a clearer presentation of our results, and we made a major revision of the manuscript.

It’s well known that neurons in the brain are also equipped with abundant estrogen-producing enzyme, aromatase, and neuron-derived estrogen is essential for memory formation. Since Punica granatum has potent anti-estrogenic effects, will Punica granatum treatment also suppress neuronal estrogen synthesis and thus impair cognitive functions? Please discuss.

Response: This aspect has been mentioned in section 5 – Limitations and Future Perspectives of Punica granatum utilization. We couldn’t find any study in the literature on this topic, but it is an interesting observation that may worth to be investigated.

Is Punica granatum also a selective estrogen modulator, like Tamoxifen which exerts distinct estrogen regulatory actions in different organs? If so, please provide evidences.

Response: This aspect has been discussed in the section Results and Discussion. Punica granatum has been demonstrated to act both on the cardiovascular system and on the brain, exerting cardio-protective and neuro-protective effects. Several scientific evidence have been refereed.

This paper concludes that Punica granatum has anti-cancer activity by a variety of mechanisms, like anti-estrogen, anti-proliferation, anti-angiogenesis, anti-inflammation, anti-metastasis et al, which is shown in Table 1. It will be more informative if the authors can summarize all these mechanisms of Punica granatum’s anti-breast cancer effects with schematic illustration in a figure.

Response: A schematic representation of the main molecular mechanisms of Punica granatum’s anti-breast cancer effects has been realized an inserted at the end of the first section, the Introduction.

Since in vivo animal studies, especially clinical studies are still lacking, it will be important if the authors can make a separate section to discuss the current limitation and future direction of Punica granatum treatment in breast cancer therapy.

Response: We introduced section 5 – Limitations and Future Perspectives of Punica granatum. Here, we highlighted the main limitations of the pomegranate use in breast cancer patients, and we also mentioned several points of interest, hoping that our study will pave the way for novel investigations in the field.

Moreover, we also made the following changes:

  • We modified the title of our paper : Pharmacological and Therapeutic Properties of Punica Granatum phytochemicals: Possible Roles in Breast cancer - A Review
  • We improved the quality of Table 1 and we introduced more studies.
  • We draw the chemical structure of the most well-known bioactive molecules from Punica granatum
  • The introduction was divided into 2 sections: 1. Introduction and 2. Punica granatum – history and chemical composition
  • We replaced the Results section summary in the abstract

Thank you for your cooperation and for your comments!

Kind regards,

Andreea Balan

Reviewer 2 Report

This paper  is a  Literature review of  bio- active compounds from pomergranates  , mainly  poly phenols , against breast-cancer.  It deals well with the many  different cited articles.As the report is aimed to be publish in Molecules the structures of several of the most active compounds should be given. 

It would be better to address the paper to pomergranate ( Punica Granatum) 

Author Response

Dear Reviewer,

We thank you for your cooperation and we appreciate you taking the time to analyze our work. Following the reviewers suggestions, we revised our paper, in order to have a clearer presentation of our results, and we made a major revision of the manuscript.

  • We modified the title of our paper : Pharmacological and Therapeutic Properties of Punica Granatum phytochemicals: Possible Roles in Breast cancer - A Review
  • We improved the quality of Table 1 and we introduced more studies.
  • We draw the chemical structure of the most well-known bioactive molecules from Punica granatum
  • The introduction was divided into 2 sections: 1. Introduction and 2. Punica granatum – history and chemical composition
  • We replaced the Results section summary in the abstract
  • We realized a schematic representation of the main molecular mechanisms of Punica granatum’s anti-breast cancer effects and inserted it at the end of the first section, the Introduction.
  • We introduced section 5 – Limitations and Future Perspectives of Punica granatum. Here, we highlighted the main limitations of the pomegranate use in breast cancer patients, and we also mentioned several points of interest, hoping that our study will pave the way for novel investigations in the field.

Reviewer 3 Report

Pharmacological and Therapeutic Properties of Punica Granatum: Possible Roles in Breast cancer - A Review

Marius Alexandru Moga, Oana Gabriela Dimienescu, Andreea Bălan, Lorena Dima, Sebastian Toma, Nicușor Florin Bîgiu, and Alexandru Blidaru

I have reviewed your manuscript for Molecules and find it acceptable for publication after major corrections.

Comments and corrections:

The topic is not novel, there has been several review articles of the topic in recent years, however the most of them are cited in this paper. Focusing on the breast cancer gives some new summary information.

Abstract: The results section summary in the abstract does not correlate well with the Results section in the paper and also in Conclusions, sentence "No
human clinical trials were conducted until now" does not give much impact.

Also it is confusing between refs [72] and [79] studies, they are both  mentioned to be human studies. [79] was mentioned only in the Table 1 and not discussed further. Please improve.

Introduction part is too long and it includes already known data which is not relevant for the breast cancer scope.

1.1 There are no compound structures included, at least the ellagtannins special for Punica granatum e.g. punicalagin and granatin A should be drawn. Also structures of single compounds in Table 1 studies should be shown.

Study should have included SciFinder database, and Google Academic database should be specified, do you mean Google Scholar? Also the time range of the selected papers should be mentioned. SciFinder-n search gave three possible relevant Chinese papers not included:

Punica granatum seed oil inhibits malignant behavior of breast cancer cellsBy: Fu, Guo-qiang; Liu, Lu; Zhang, Lei; Gao, Yuan; Xu, Xiao-na; Xie, Feng; Wang, Feng Junshi Yixue (2015), 39(6), 438-442 | Language: Chinese

Influence of pomegranate peel ellagic acid on the immune function of mice with 4T1 breast cancer. By: Zhang, Yumei; Wang, Jiaxiao Xiandai Zhongxiyi Jiehe Zazhi (2014), 23(15), 1597-1599, 1602 | Language: Chinese

Effects of extract of pomegranate peel on proliferation and apoptosis of breast cancer cell line mda-mb-231, By: Zhang, Xun; Wu, Ai-ping; Zhang, Yong-hong Tianjin Zhongyiyao Daxue Xuebao (2012), 31(4), 214-217 | Language: Chinese

I would have been interested to read more discussion about pomegranate constituents’ binding to estrogen receptor beta (especially isoflavones) and role of that in breast cancer studies. Also critical view taking into account the possibility of polyphenols acting as PAINS (Pan assay interference compounds) could have been worth to study.

Table 1. is not well readable and fluent. The order of row is not logical. Pomegranate extract is not a correct title for column if there are also single compounds like punicic acid. Results column is incoherent and the column is too narrow. First entry Kim et al. the cell types are bold but not in the others. Please modify the table visually better and positioned and more clear. Table 1 is not referred anywhere in the text and the most results there are not commented at all. Please improve.

Figures 1 and 2. Punicallin should be Punicalin

Figure 3. Sterols should be Sterols and steroids

Title: Please replace Cranatum with cranatum

Line 188: Please replace 17β-hydroxysteorid with 17β-hydroxysteroid

Lines 145 and 146: Please replace potasium with potassium

Line 318 and 323: Please replace CO2 with CO2

Line 357: Is the book reference [72] correct for this study? Chapter and its authors should be stated in the reference

References: Check the correct style for the references. Please replace Octomber with October

General: Compound names should not be written with capital in sentences e.g tamoxifen

Author Response

Dear Sir or Madam,

We thank you for your cooperation and we appreciate you taking the time to analyze our work. Following your comments and suggestions, we revised our paper, in order to have a clearer presentation of our results, and we made a major revision of the manuscript.

Abstract: The results section summary in the abstract does not correlate well with the Results section in the paper and also in Conclusions, sentence "No human clinical trials were conducted until now" does not give much impact.

Response: We replaced the results and conclusion sections summary in the abstract

Introduction part is too long and it includes already known data which is not relevant for the breast cancer scope.

Response: We divided the Introduction into 2 sections: Section 1 Introduction and Section 2 Punica granatum – history and chemical composition

There are no compound structures included, at least the ellagtannins special for Punica granatum e.g. punicalagin and granatin A should be drawn. Also structures of single compounds in Table 1 studies should be shown.

Response: We draw the chemical structure of the most well-known compounds of pomegranate

Study should have included SciFinder database, and Google Academic database should be specified, do you mean Google Scholar? Also the time range of the selected papers should be mentioned. SciFinder-n search gave three possible relevant Chinese papers not included

Response: -     We mentioned the time range of the selected papers

  • We have not access to the SciFinder n database, and even if we would have had access, these three articles are written in Chinese language. Articles written in other languages than English represented an exclusion criteria.

I would have been interested to read more discussion about pomegranate constituents’ binding to estrogen receptor beta (especially isoflavones) and role of that in breast cancer studies. Also critical view taking into account the possibility of polyphenols acting as PAINS (Pan assay interference compounds) could have been worth to study.

Response: More discussion about pomegranate constituents’ binding to estrogen receptor beta (genistein) have been introduced in the Results and Discussion section. The possibility of polyphenols acting as PAINS was mentioned in Section 5, but from our knowledge, there is no study in the literature on this topic until present.

Table 1. is not well readable and fluent. The order of row is not logical. Pomegranate extract is not a correct title for column if there are also single compounds like punicic acid. Results column is incoherent and the column is too narrow. First entry Kim et al. the cell types are bold but not in the others. Please modify the table visually better and positioned and more clear. Table 1 is not referred anywhere in the text and the most results there are not commented at all. Please improve.

Response: Table 1 was improved; 2 more preclinical studies were introduced in the table. All the studies introduced in the table were discussed in section 4, Results and Discussion.

Figures 1 and 2. Punicallin should be Punicalin

Response: Punicallin was replaced

Figure 3. Sterols should be Sterols and steroids

Response: we made this modification

Title: Please replace Granatum with granatum

Response: Granatum was replaced by granatum

Line 188: Please replace 17β-hydroxysteorid with 17β-hydroxysteroid

Response: we made this modification

Lines 145 and 146: Please replace potasium with potassium

Response: we made this modification

Line 318 and 323: Please replace CO2 with CO2

Response: we made this modification

Line 357: Is the book reference [72] correct for this study? Chapter and its authors should be stated in the reference

Response: we made this modification

References: Check the correct style for the references. Please replace Octomber with October

Response: we made this modification

General: Compound names should not be written with capital in sentences e.g tamoxifen

Response: we made this modification

Thank you very much for your collaboration!

Kind regards,

Andreea Balan

Round 2

Reviewer 3 Report

I have rereviewed your manuscript for Molecules and find it well improved and acceptable for publication after minor corrections.

Please clean up the structures in Figure 5 and preferably draw them with ChemDraw, some bond angles and rings are not ok. And there are weird squares at oxygens in a), c) and d). You could also make the figure more compact by making the structures a bit smaller. Compounds should also be numbered and numbers used in text.

Table 1. entry Mandal 2017 adjust the lines in the last column

Author Response

Dear Reviewer, 

We made the requested revisions. The chemical structure of the most important biomolecules from pomegranate was drawn using KingDrawn. Also, the last column of the entry Mandal 2017 has been adjusted. 

Thank you very much for your collaboration!

Kind regards, 

Andreea Balan